# Effect of Gut Microbiota on Blood Cholesterol: A Review on Mechanisms

**DOI:** 10.3390/foods12234308

**Published:** 2023-11-29

**Authors:** Chuanling Deng, Jingjin Pan, Hanyue Zhu, Zhen-Yu Chen

**Affiliations:** 1School of Food Science and Engineering/National Technical Center (Foshan) for Quality Control of Famous and Special Agricultural Products (CAQS-GAP-KZZX043), Foshan University, Foshan 528011, China; dchuanling@126.com (C.D.); panjingjin@foxmail.com (J.P.); 2School of Life Sciences, The Chinese University of Hong Kong, Shatin, NT, Hong Kong, China

**Keywords:** gut microbiota, cholesterol-lowing mechanisms, natural functional ingredients

## Abstract

The gut microbiota serves as a pivotal mediator between diet and human health. Emerging evidence has shown that the gut microbiota may play an important role in cholesterol metabolism. In this review, we delve into five possible mechanisms by which the gut microbiota may influence cholesterol metabolism: (1) the gut microbiota changes the ratio of free bile acids to conjugated bile acids, with the former being eliminated into feces and the latter being reabsorbed back into the liver; (2) the gut microbiota can ferment dietary fiber to produce short-chain fatty acids (SCFAs) which are absorbed and reach the liver where SCFAs inhibit cholesterol synthesis; (3) the gut microbiota can regulate the expression of some genes related to cholesterol metabolism through their metabolites; (4) the gut microbiota can convert cholesterol to coprostanol, with the latter having a very low absorption rate; and (5) the gut microbiota could reduce blood cholesterol by inhibiting the production of lipopolysaccharides (LPS), which increases cholesterol synthesis and raises blood cholesterol. In addition, this review will explore the natural constituents in foods with potential roles in cholesterol regulation, mainly through their interactions with the gut microbiota. These include polysaccharides, polyphenolic entities, polyunsaturated fatty acids, phytosterols, and dicaffeoylquinic acid. These findings will provide a scientific foundation for targeting hypercholesterolemia and cardiovascular diseases through the modulation of the gut microbiota.

## 1. Introduction

The intestinal microbial structure embodies a complex bacterial consortium encompassing over 35,000 distinct bacterial species [1]. This consortium predominantly consists of four phyla, namely, *Firmicutes*, *Bacteroides*, *Actinobacteria*, and *Proteobacteria*, with *Firmicutes* and *Bacteroides* dominating, accounting for a substantial 90% of the total species count [2]. Intestinal microbes could profoundly impact host health, influencing not only host’s metabolic processes, including the absorption of nutrients, but also the metabolism of detrimental substances [3]. The composition and functionality of the gut microbiota are subjected to modulation by many factors, encapsulating both internal elements and external determinants such as genetics, age, diet, lifestyle, and medications [4]. Notably, diet is one of most pivotal factors underpinning alterations in the intestinal microbial structure [5,6]. Functional constituents in diets can regulate the growth and metabolic activities of the gut microbiota, thereby influencing the microbial composition. It is imperative to highlight that discernible disparities exist in the genetic content of intestinal microbes between adults and infants, reflecting the divergent intestinal functional needs at different life stages [7]. Through the production of an array of metabolic products such as short-chain fatty acids (SCFAs), branched-chain fatty acids (BCFAs), bile salt hydrolase (BSH), and lipopolysaccharides (LPS), the gut microbiota partakes in and regulates the host’s metabolism. Notably, SCFAs and BCFAs play a crucial role in this process and furnish vital nutritional sources for intestinal cells [8]. SCFAs can stimulate the proliferation and differentiation of intestinal epithelial cells, contributing to the maintenance of the mineral balance and the absorption of iron, calcium, and magnesium [9]. Prevalent SCFAs include acetate (Ac), propionate (Pr), and butyrate (Bu), whereas, in the colon, others such as valerate (Va), caproate (Ca), and isobutyrate are relatively low, constituting approximately 5–10% of the total SCFAs [10]. LPS is identified as an endotoxin derived from Gram-negative bacteria. Elevated LPS levels are associated with some metabolic diseases, inflammation, the infiltration of adipose macrophages, endothelial cell apoptosis, fatty-liver-related diseases, and insulin resistance [11,12,13]. Additionally, intestinal microbes can produce BSH, an enzyme capable of hydrolyzing N-acylamide bonds, facilitating the release of taurine or glycine (Gly) through the hydrolysis of bile salts [14].

Cholesterol is a sterol synthesized endogenously by animals, serving as an indispensable component of cell membranes. Beyond its structural role, it acts as a signaling molecule in various biological processes, including cellular transport and neurotransmission. It also functions as a precursor for synthesizing vitamin D, steroid hormones (such as progesterone and estrogen), and bile acids [15]. Due to its low water solubility, cholesterol predominantly circulates within the body through lipoproteins [16]. Depending on the density, size, and composition of these lipoproteins, they are classified into chylomicrons (CMs), very-low-density lipoproteins (VLDL), intermediate-density lipoproteins (IDL), low-density lipoproteins (LDL), and high-density lipoproteins (HDL) [17]. The homeostasis of cholesterol plays a vital role in physiological functions. It has been long known that an elevated serum cholesterol level, or hypercholesterolemia, is a principal cause of atherosclerosis and coronary heart disease [18]. Conversely, an overly reduced cholesterol level may also pose a health risk, including an increased susceptibility to hemorrhagic stroke and a correlation with higher mortality rates due to late-stage heart failure [19,20].

Many recent studies have unveiled a significant correlation between gut microbiota dysbiosis and cholesterol metabolism. In this regard, a profound exploration into the influence exerted by the intestinal microbiota on cholesterol metabolism is of paramount importance. In general, the gut microbiota can engage in cholesterol metabolism through the following possible pathways: modulating the ratio of free to conjugated bile acids [21]; enhancing the abundance of the SCFA-producing microbiota to increase the concentration of SCFAs in the intestinal lumen [22]; and regulating the expression of cholesterol-metabolism-related genes, epitomized by the activity of Lactobacillus strains [23]. In addition, the gut microbiota facilitates the conversion of cholesterol into fecal neutral sterols for excretion, while it concurrently reduces the production of LPS [24,25]. Therefore, optimizing the abundance of beneficial bacteria in the colon represents one of the primary strategies for cholesterol reduction. Currently, probiotics and prebiotics as health supplements are widely advocated [26]. Compared with the direct intake of supplements, bioactive compounds in natural foods boast higher bioavailability, potentially conferring additional health benefits [27]. For instance, polysaccharides, ubiquitously present in various plants, can be hydrolyzed and fermented to produce SCFAs by the intestinal microbiota [28]. Polyphenolic compounds, phytosterols (PS), and polyunsaturated fatty acids (PUFAs) are noted for their diversity and abundance, with the capacity to stimulate the growth of a beneficial microbiota [29,30]. Recent studies have highlighted the potential of dicaffeoylquinic acid (DCQA), a functional component found in *Ilex kudingcha*, in promoting the growth of SCFA-producing bacteria [31]. Beyond cholesterol modulation, these natural bioactive components also exhibit some immunomodulatory and anti-inflammatory activities [32], playing a positive role in preventing and treating various diseases. Integrating these natural functional components into strategies for modulating the intestinal microbiota to regulate cholesterol metabolism opens a new avenue for therapeutic intervention.

## 2. Pathways of Cholesterol Metabolism

Epidemiological studies have evidenced a compelling association between elevated cholesterol levels and cardiovascular diseases (CVDs), the latter being a predominant cause of mortality and disability in developed countries, projected by health organizations to persist until 2030 [33,34]. Cholesterol in humans primarily derives from two sources: endogenous cholesterol synthesized in the liver and intestine, and exogenous cholesterol acquired through the consumption of animal-derived foods [35]. Cholesterol homeostasis is vital for the physiological balance among hepatic cholesterol synthesis, absorption, transport, and biliary excretion (Figure 1).

### 2.1. Cholesterol Synthesis

The liver is pivotal in cholesterol homeostasis, spearheading its synthesis and conversion into bile acids [36]. Cholesterol biosynthesis involves a cascade of enzymatic reactions [37]. Initially, acetyl-CoA is transformed into mevalonate (MVA) though several reactions. This transformation has two acetyl-CoA molecules merging into acetoacetyl-CoA, which then combines with another acetyl-CoA, facilitated by HMG-CoA synthase (HMG-CoA-S), to form HMG-CoA. This compound is subsequently reduced to MVA by HMG-CoA reductase (HMG-CoA-R), the rate-limiting enzyme crucial for averting excessive cholesterol synthesis and accumulation [38,39,40]. MVA then morphs into isopentenyl pyrophosphate (IPP), with mevalonate kinase (MK), phosphomevalonate kinase (PMK), and mevalonate diphosphate decarboxylase (MDD) governing this transition [41,42]. Through condensation, IPP evolves into squalene (SQ), a 30-carbon precursor to all steroids. SQ is then oxidized to 2,3-oxidosqualene (OS) by squalene monooxygenase (SM) and cyclized to lanosterol by oxidosqualene cyclase-lanosterol synthase (OSC) [43,44,45]. Lanosterol ultimately transforms into cholesterol after a series of complex reactions [46]. The lanosterol-to-cholesterol conversion stage is a complex process with numerous enzymes at play and still requires further elucidation regarding its structure and mechanisms.

### 2.2. Cholesterol Absorption

Cholesterol absorption in the small intestines is an intricately regulated physiological process governed at the cellular level by a series of proteins. Cholesterol within the intestinal tract primarily originates from dietary intake, biliary secretion, and the intestinal mucosal epithelium. Western diets are estimated to contribute approximately 300–500 mg of cholesterol daily, while the bile provides 800–1200 mg, and the intestinal mucosal epithelium adds around 300 mg [47]. Cholesterol absorption begins in the stomach, forming micelles after being emulsified by bile acids in the small intestine. It is important to note that only non-esterified cholesterol can form these micelles [48]. These micelles subsequently interact with the Niemann-Pick C1-like 1 protein (NPC1L1), a pivotal transporter in cholesterol absorption processes, facilitating the micelles’ transport into the intestinal epithelial cells [49]. NPC1L1 predominantly localizes at the apical membrane of enterocytes in the small intestine [50]. Entering the intestinal epithelial cells, cholesterol is esterified by acyl-coenzyme A: cholesterol acyltransferases 2 (ACAT2) in the endoplasmic reticulum, forming cholesterol ester (CE). Subsequently, microsomal triglyceride transfer protein (MTP) transfers CE into chylomicrons (CMs), which enter the lymphatic system and the bloodstream, and are transported to the liver [51,52]. Two forms of ACAT enzymes, namely, ACAT1 and ACAT2, have been identified in mammals. While ACAT1 is ubiquitously expressed in various tissues, ACAT2 is predominantly found in intestinal epithelial cells and hepatocytes [53]. MTP is a lipid transfer protein responsible for transporting CE from the endoplasmic reticulum to nascent apoB lipoproteins, further facilitating the assembly of CM [54]. ApoB lipoproteins primarily mediate the transportation and metabolism of cholesterol and triglycerides [55]. Non-esterified cholesterol, on the other hand, is transported back into the intestinal lumen by ATP-binding cassette transporters G5 and G8 (ABCG5/8) [51]. These transporters, functioning as heterodimers, are prominently expressed in the microvilli of intestinal cells and the canalicular membrane of hepatocytes, playing a collective role in cholesterol excretion [56].

### 2.3. Cholesterol Excretion

Human body excretes approximately one gram of cholesterol daily, half of which is transformed into bile acids (BAs) and eliminated through feces [57]. At the same time, the remainder exists unesterified within fecal matter. In the liver, cholesterol 7α-hydroxylase (CYP7A1) and sterol 27-hydroxylase (CYP27A1) catalyze the 7-α-hydroxylation and 27-hydroxylation of cholesterol, respectively, further synthesizing primary bile acids, cholic acid (CA), and chenodeoxycholic acid (CDCA). These primary bile acids accumulate in the bile, conjugated with glycine (Gly) or taurine [58,59,60]. Some primary bile acids undergo deconjugation and 7α-dehydroxylation in the intestinal tract, generating secondary bile acids, namely, deoxycholic acid (DCA) and lithocholic acid (LCA) [61,62]. Both primary and secondary bile acids are partially absorbed in the ileum and returned to the liver through the portal venous system [57]. Due to its insolubility, LCA is generally poorly reabsorbed [63]. Bile acids that are not absorbed are excreted as fecal acidic sterols [64]. Approximately 3–5 g of bile acids circulate within the intestine multiple times (between 6–10 cycles), a process under sophisticated feedback regulation [65]. Within this regulatory framework, CYP7A1, acting as the rate-limiting enzyme for bile acid biosynthesis, has its activity negatively modulated by the nuclear bile acid receptor, Farnesoid X receptor (FXR). When the bile acid pool within the enterohepatic circulation increases, FXR is activated, thereby inhibiting the transcriptional activity of the CYP7A1 gene [66]. CDCA is crucial in activating FXR, its most potent ligand [67].

A second pathway involves the excretion of cholesterol by intestinal cells, manifesting as fecal-neutral sterols (FNSs) [68]. Cholesterol that was unabsorbed by the small intestine is transported to the intestinal lumen by ABCG5/8, eventually excreted as fecal-neutral sterols [51]. This process is positively regulated by Liver X receptor α (LXRα), a principal regulator participating in the mRNA expression of ABCG5/8 [69]. Additionally, ABCG5/8 facilitates the secretion of cholesterol and phytosterol into the bile [70]. The overexpression of ABCG5/8 reduces the absorption of dietary cholesterol [71].

## 3. Mechanisms through Which Gut Microbiota Influences Cholesterol Metabolism

In recent years, abundant research has centered around understanding how gut microbiota communities influence human health. One emerging piece of evidence is that the gut microbiota can affect cholesterol metabolism. These microbes engage in cholesterol metabolism through various mechanisms to reduce plasma cholesterol levels. It has been elucidated that Lactobacillus alone embodies multiple distinct mechanisms for cholesterol removal [72]. In general, the gut microbiota achieves cholesterol reduction through the following mechanisms: transforming complex non-digestible polysaccahrides into monosaccharides and fermenting them to produce beneficial SCFAs [73]; generating BSH, which facilitates the deconjugation of conjugated bile acids and releases free bile acids [74]; participating in the regulation of gene expressions associated with cholesterol metabolism [75]; promoting the conversion of cholesterol into fecal sterols [76]; and influencing the production of LPS, which affects cholesterol levels [25].

### 3.1. Participation of Gut Microbiota in Modifying Conjugated Bile Acids

Some gut microbes can produce BSH and hydrolyze the conjugated bile acids into free bile acids, thus increasing the ratio of free bile acids to conjugated bile acids, leading to a greater excretion of bile acids, and resulting in a smaller pool of both cholesterol and bile acids [14]. This is because the free bile acids are mostly excreted into feces, whereas the latter ones are reabsorbed back into the liver [64]. A portion of hepatic cholesterol undergoes conversion into BA. In conjunction with glycine (Gly) and taurine, these BA molecules form conjugated bile acids (C-BAs), which enter the small intestine. Lactic acid bacteria (LAB) secrete BSH within the intestinal environment. Under the catalytic influence of BSH, C-BA deconjugates, giving rise to free BA, Gly, and Taurine. The generated free BA is subsequently excreted from the body, while the remaining C-BA recirculates to the liver via the portal vein (Figure 2). In addition, the liver plays a pivotal role in maintaining cholesterol homeostasis and orchestrating cholesterol synthesis and its conversion into bile acids for elimination [77]. The conjugation process results in a reduced acid dissociation constant (pKa) and the complete ionization of these acids, which exist in the form of anions [78,79,80]. Bile acids undergo various biochemical modifications in the human large intestine, including deconjugation, 7α/β-dehydroxylation, and epimerization [21]. Deconjugation is achieved through the enzymatic hydrolysis of the C24 N-acylamide bond that links bile acids with their conjugated amino acids. These deconjugated primary bile acids function as signaling molecules, reflecting the body’s total bile acid levels and increasing the concentrations of cholic acid (CA) and chenodeoxycholic acid (CDCA). Furthermore, glycine and taurine released during deconjugation serve as nutrient sources for the intestinal microbiota [81]. Approximately 26.03% of the total bacterial population in the large intestine exhibits BSH activity [82]. BSH with deconjugation capabilities primarily resides in Gram-positive bacteria, including *Bifidobacterium*, *Lactobacillus*, *Clostridium*, *Enterococcus*, and *Listeria* [83,84,85,86,87,88]. Nonetheless, BSH activity is not exclusive to Gram-positive bacteria; Gram-negative bacteria like *Stenotrophomonas*, *Bacteroides*, and *Brucella* also exhibit BSH activity [89,90,91]. All BSH deconjugation reactions depend on the hydrolysis of the N-acylamide bond, releasing taurine or glycine, with the reaction exhibiting maximum activity in neutral or mildly acidic environments (pH 5–7), with an optimal pH of approximately 6 [83,92].

Scientists have identified a gene cluster in *Clostridium scindens* (*C. scindens*), the bai operon, crucially implicated in bile acid dehydroxylation. This operon encodes for a multitude of enzymes essential for the dehydroxylation process [93]. The baiG gene within this cluster encodes bile acid transport proteins, facilitating the uptake of CA by bacterial strains and transporting CDCA and DCA [94]. Under the influence of baiB, bile acids are oxidized to form cholyl-coenzyme A (CoA), which is then further oxidized by baiA2 to produce 3-oxo-cholyl-CoA. Subsequently, baiCD catalyzes the oxidation of 3-oxo-cholyl-CoA to form 3-oxo-Δ4-cholyl-CoA. Then, baiF transfers CoA from 3-oxo-Δ4-cholyl-CoA to CA, resulting in the formation of 3-oxo-Δ4-CA and cholyl-CoA [95]. This 3-oxo-Δ4-CA, under the action of baiE, undergoes dehydroxylation to yield 3-oxo-Δ4,6-DCA, the rate-limiting step in the process [93]. Following this, continuous activity by baiN on 3-oxo-Δ4,6-DCA produces 3-oxo-DCA, which is then converted to DCA through the combined effort of baiO and baiA2, occurring at the C7 position, known as 7α-dehydroxylation [96,97]. 7β-dehydroxylation occurs similarly, with the primary distinction being the utilization of baiH instead of baiCD for the oxidation at the C4 position, with the activity of the 7β-dehydratase enzyme possibly serving as the rate-limiting step for 7β-dehydroxylation. Currently, bacteria such as *C. scindens*, *C. hylemonae*, *C. perfringens*, and *P. hiranonis* have been observed to produce enzymes capable of facilitating the 3α-dehydrogenation of hydroxysteroids, a crucial step within the 7α-dehydroxylation pathway [98,99,100].

In the metabolic pathways of bile acids, positional isomerization, a significant biochemical process, gives rise to many functional derivatives. This mechanism primarily hinges on the action of location-specific hydroxysteroid dehydrogenases (HSDHs), such as 7α-HSDH, which oxidize hydroxyl groups [101]. This process is then followed by the reduction facilitated by another location-specific hydroxysteroid dehydrogenase, 7β-HSDH. Enzymes analogous to these include 3α/β-HSDH and 12α/β-HSDH [102,103]. Through positional isomerization, CA can be transformed into various derivatives, including ursodeoxycholic acid (UCA), 12-epi-cholic acid (12-ECA), or iso-cholic acid (iCA). Similarly, CDCA can undergo isomerization to yield ursodeoxycholic acid (UDCA) or iso-chenodeoxycholic acid (iCDCA) [81]. These isomerization reactions enhance the diversity and metabolism of bile acids and further promote cholesterol metabolism. Existing studies corroborate that specific intestinal micro-organisms, such as *Clostridium baratii*, can isomerize CDCA to UDCA [104]. In addition, several other gut microbes—including *Ruminococcus*, *Clostridium*, *Stenotrophomonas maltophilia*, and *Collinsella aerofaciens*—have been verified to generate UDCA through the activity of 7α/β-HSDH.

### 3.2. Microbial Production of SCFAs and Their Effects on Cholesterol Metabolism

SCFAs in intestines are pivotal in sustaining human health. Specifically, various bacteria, including *Alloprevotella*, *Bacteroides*, *Clostridium*, *Eubacterium*, *Faecalibacterium*, and *Roseburia*, are known to produce these beneficial SCFAs, with Bu being a prominent member [105]. Bu has demonstrated its therapeutic potential in various diseases, including gastrointestinal disorders, the regulation of carbohydrate metabolism, and an improvement in obesity [106]. Further research indicates a connection between Bu and cholesterol metabolism. Previous studies have revealed that Bu can reduce the serum low-density lipoprotein cholesterol (LDL-C) level, a crucial risk factor for cardiovascular diseases [107]. Currently, statins are the preferred treatment for lowering LDL-C, primarily by inhibiting HMG-CoA-R, consequently upregulating the expression of LDL receptors (LDLRs), which enhances LDL uptake from the circulation, ultimately reducing LDL-C levels in the plasma [108,109]. Furthermore, the sterol-regulatory element binding protein-2, a key regulator of cholesterol metabolism and homeostasis, increases LDLR expression upon activation [110]. SCFAs, such as Bu as exemplified in Figure 3, generated by the gut microbiota participate in cholesterol metabolism through two distinct pathways. Firstly, Bu acts to inhibit the expression of HMG-CoA-R, thus further suppressing cholesterol synthesis, ultimately leading to reduced cholesterol levels. Secondly, Bu influences the activity of SREBP-2, thereby promoting the expression of LDL-R. The upregulation of LDL-R expression accelerates the uptake of LDL from the bloodstream, ultimately resulting in lowered levels of LDL-C (Figure 3) [22]. It is noteworthy that the mechanism of Bu differs significantly from that of statins.

Beyond Bu, other SCFAs have also exhibited cholesterol-lowering properties. For instance, it has been shown that injecting Pr into the ceca of rats fed with a casein-based diet results in a noticeable reduction in plasma cholesterol levels [111]. Furthermore, Ac could inhibit hepatic lipid synthesis and reduce TC and TG levels in mice given a high-fat diet [112]. The supplementation of SCFAs with two to four carbons into the diet reduces blood cholesterol in hamsters [113]. As other SCFAs like valerate (Va), caproate (Ca), and isobutyrate are quantitatively very low in the colon, no sufficient research data support their cholesterol-lowering activities, and this warrants further investigation.

### 3.3. Gene Expression Involvement of Lactobacillus in Cholesterol Metabolism

Research has been shown that lactic acid bacteria (LAB) can remarkably mitigate cholesterol levels via several mechanisms, including assimilation, absorption, and co-precipitation [114,115]. One study has unveiled that *Lactobacillus fermentum SM-7* can absorb and co-precipitate up to 38.5% of cholesterol and assimilate an additional 60% [116]. LAB also plays a crucial role in cholesterol reduction by regulating the gene expression of enzymes involved in cholesterol synthesis, absorption, and excretion. The phosphorylation activity of AMPK governs various regulators and transcription factors implicated in lipid metabolism [117]. In this regard, *Lactiplantibacillus plantarum DR7* can downregulate the mRNA of HMG-CoA-R by mediating AMPK phosphorylation, subsequently lowering cholesterol levels [118]. Moreover, approximately 50% of daily dietary cholesterol is absorbed through the intestines, with the remainder being excreted through feces [23]. Dietary cholesterol requires specific binding with NPC1L1 in intestinal epithelial cells for absorption, whereas it requires ABCG5/G8 to shuttle cholesterol back to the lumen of the intestine for elimination [119,120]. Noteworthy is the discovery that LAB, through the activation of PPAR and LXR, influences the expression of ABCG5/G8 and NPC1L1, playing a significant role in cholesterol excretion and absorption processes [121].

SREBPs, expressed principally in the liver, encompass three subtypes: SREBP-1a, SREBP-1c, and SREBP-2 [122]. SREBP-1a can effectively activate all SREBP-responsive genes, inclusive of those involved in the synthesis of cholesterol, fatty acids, and triglycerides. On the other hand, SREBP-1c prioritizes the transcription activation of genes necessary for fatty acid synthesis without activating cholesterol-related genes. In contrast, SREBP-2 primarily activates LDL-R genes and those requisite for cholesterol synthesis [123]. Both *Lactobacillus plantarum NCU116* and *L. brevis SBC8803* have been demonstrated to impede cholesterol accumulation by influencing SREBP expressions, ultimately reducing the cholesterol concentration [124,125]. Lastly, CYP7A1, an enzyme facilitating bile acid synthesis, is integral in maintaining mammal cholesterol homeostasis [126]. Notably, LXRα and FXR act as positive and negative regulators in cholesterol metabolism, modulating the expression of CYP7A1 mRNA [127]. It has been shown that *Lactobacillus plantarum H6* could increase bile acid synthesis and CYP7A1 expression by suppressing FXR target gene expression [128]. Furthermore, the transcription of CYP7A1 is negatively regulated by FGF15 signaling. The research conducted by Kim et al. found that *Lactobacillus rhamnosus GG* could suppress FGF15 expression, promoting an increase in CYP7A1 expression in the liver, and reducing total cholesterol levels [129].

### 3.4. Probiotic Conversion of Cholesterol to Coprostanol

Coprostanol possesses a distinctive cis A/B ring configuration in its chemical structure, prompting the shift of 3-OH from the axial to the equatorial position. This unique structural adjustment potentially hinders the incorporation of coprostanol into mucosal cells, consequently limiting its absorption in the intestines [130]. Hence, it is perceived that this transformative process is an effective approach to reduce plasma TC levels because the gut microbiota could transform cholesterol to coprostanol [24,131]. It transpires that the microbial conversion of cholesterol to coprostanol in the intestine is mediated by three primary pathways. The process of converting cholesterol into coprostanol can be categorized into one direct pathway and two indirect pathways. In the direct pathway, cholesterol undergoes reduction, specifically targeting the 5–6 double bond, resulting in the formation of coprostanol. The first indirect pathway involves a series of reactions catalyzed by various enzymes, including cholesterol oxidase, encompassing oxidation, isomerization, and reduction processes that ultimately lead to the production of coprostanol. The second indirect pathway is through the allocholesterol pathway, distinct from the standard cholesterol pathway, leading to the reduction of cholesterol into coprostanol. (Figure 4). Two of these are indirect: Initially, cholesterol is oxidized to intermediary 5-alpha-cholestan-3-one under the influence of cholesterol oxidase; subsequently, 5-alpha-cholestan-3-one undergoes isomerization to form 4-cholesten-3-one, which is then reduced to coprostanone; and, finally, coprostanone is further reduced to coprostanol [132,133]. Another route involves the isomerization of cholesterol to allocholesterol, followed by the reduction of allocholesterol to coprostanol [134,135,136]. Additionally, a direct pathway exists, where cholesterol is transformed into coprostanol through the direct reduction of the 5–6 double bond; however, this pathway has been less extensively researched [137,138]. In this regard, *Eubacterium coprostanoligenes ATCC 51222* could convert 90% of cholesterol into coprostanol in the medium [133] and *E. ATCC 21408* could directly convert cholesterol into coprostanol through intermediary steps involving 4-cholesten-3-one and coprostanone [134]. Furthermore, a mixed culture of *Lactobacillus acidophilus 43121*, *Lactobacillus casei*, and *Bifidobacterium* could reduce TC levels and augment coprostanol excretion [130,139]. Studies have shown that probiotics like *Bifidobacterium*, *Lactobacillus*, and *Clostridium* can convert cholesterol to coprostanol under in vitro conditions [140,141]. Although the probiotic-mediated conversion of cholesterol to coprostanol is substantiated as an effective cholesterol-lowering mechanism, the challenge still exists, including the identification of specific microbial strains and enzymes involved in the process. Given the high oxygen sensitivity of these micro-organisms, the number of strains successfully isolated for the purpose of cholesterol reduction is considerably limited. Therefore, future work necessitates a continuation in the research on microbial strain isolation and the comprehensive genomic analysis of these strains to elucidate their precise roles in cholesterol reduction.

### 3.5. Lipopolysaccharides’ Involvement in Cholesterol Metabolism

A healthy gut microbiota is associated with a low production of LPS. In general, LPS is a component embedded within the outer membrane of Gram-negative bacteria [142]. Existing literature illustrates LPS’s dynamic engagement with lipids in the bloodstream through various mechanisms. Earlier studies had shown that LPS could increase LDL cholesterol, whereas it decreased HDL cholesterol, presumably by promoting HMG-CoA reductase [143]. In addition, the concentration of triacylglycerols in the blood can also be modulated through distinct pathways activated by LPS. One study has delineated that lower doses of LPS could stimulate the hepatic synthesis of very-low-density lipoprotein (VLDL), whereas higher doses inhibited lipoprotein metabolic degradation [144]. It is most likely that the gut microbiota modulates plasma cholesterol and triacylglycerol partially by affecting the production of LPS.

LPS is also proinflammatory. Numerous studies have demonstrated that LPS possesses a significant binding affinity with TC. Upon binding, these LPS-TC complexes are transported via the lymphatic system, potentially inducing inflammatory responses [145]. Further research indicates that LPS can activate toll-like receptors 4 and 9 (TLR4 and TLR9), subsequently triggering the NLR family pyrin domain-containing 3 (NLRP3) inflammasome, a process believed to be involved in the fibrotic progression of non-alcoholic fatty liver disease (NAFLD) [146]. Studies by Yoshida et al. discovered that strains *Bacteroides vulgatus* and *Bacteroides dorei* could reduce the concentration of LPS produced by the intestinal microbiota [25]. This function might positively contribute to the alleviation of atherosclerosis [25]. However, research exploring cholesterol reduction through the modulation of intestinal LPS levels remains scant.

## 4. Natural Functional Constituents Influencing Gut Microbiota in the Regulation of Cholesterol Metabolism

Natural functional components, encompassing indigestible polysaccharides, phenolic compounds, unsaturated fatty acids, and phytosterols, have manifested as being capable of fostering the proliferation of probiotics in the intestine. They fortify the human immune system by activating or modulating immune cells and responses. Furthermore, these functional constituents can be deployed as adjunctive measures to prevent cardiovascular diseases and certain inflammatory conditions. Regarding their effect on plasma cholesterol mediated by the gut microbiota, these constituents primarily exert their influence by enhancing the proliferation of SCFA-producing strains, modulating strains involved in cholesterol metabolism, promoting BSH-producing strains, and facilitating the conversion of cholesterol to coprostanol.

### 4.1. Indigestible Polysaccharides

Indigestible polysaccharides, abundant natural prebiotics, positively sway the gut microbiota and their metabolism. They not only adjust the microbiota composition but also promote the growth of beneficial bacteria. Upon consumption, indigestible polysaccharides could reach the large colon where they are fermented by the gut microbiota, producing SCFAs, primarily by *Bacteroides* and *Firmicutes* members. *Bacteroides Thetaiotaomicron* produce propionate (Pr) and acetate (Ac), which are subsequently transformed into butyrate by *Eubacterium rectale* [147]. Furthermore, an increase in butyrate production is observed with *F. prausnitzii* [148]. It is imperative to highlight that butyrate can be produced by the intestinal microflora through fiber fermentation or the Wood–Ljungdahl pathway [149]. Studies corroborate the efficacy of SCFAs in the reduction of plasma cholesterol. Seaweed polysaccharides have been demonstrated to be capable of alleviating gut microbiota dysbiosis and reducing cholesterol [150]. For instance, polysaccharides derived from red algae could increase the production of SCFAs, favorably modulate the gut microbiota, and reduce cholesterol [151]. Additionally, polysaccharides from Porphyra could alleviate the diet-induced intestinal dysbiosis by enhancing the population of *Eubacterium xylanophilum*, a known butyrate producer [152]. Alginate oligosaccharides derived from brown algae could elevate the BSH activity and enhance the CYP7A1 activity, facilitating bile acid synthesis and cholesterol reduction [153]. BSH is predominantly produced in the intestine by a consortium of bacteria, including *Bacteroides*, *Bifidobacterium*, *Clostridium*, *Enterobacter*, *Enterococcus*, and *Lactobacillus* [154]. Beyond seaweeds, recent findings emphasize the ability of polysaccharides from edible fungi to regulate the gut microbiota composition. Research has shown that polysaccharides from Auricularia auricula could stimulate the growth of SCFA-producing bacteria like *Oscillibacter* and *Lactobacillus*, ultimately enhancing the production of intestinal SCFAs and thereby reducing cholesterol [155]. Similar functions have been also observed with polysaccharides derived from mushrooms and *Pleurotus eryngii* [156,157].

### 4.2. Polyphenolic Compounds

Polyphenolic compounds are significantly active entities characterized by antioxidant and anti-inflammatory properties found extensively within various plants. These compounds, classified into phenolic acids, flavonoids, tannins, and lignans based on their unique compositional and structural characteristics, play vital roles in human health [29]. Tea leaves are rich sources of various polyphenolic compounds, including catechins, epicatechins, and quercetin (QR) [158]. Tzounis et al. discovered that catechins enhance the proliferation of the *Blautia coccoides-Eubacterium rectale* group and *Bifidobacterium* in the gut, with the former being recognized for increasing the concentration of SCFAs within the intestinal environment [159,160]. Furthermore, their research unveils the significant role of flavanols in cocoa: these compounds not only elevate the levels of *Bifidobacterium* and *Lactobacillus* in the gut but also inhibit the growth of certain pathogenic micro-organisms [161]. Moreover, red wine is a substantial source of polyphenolic compounds, including but not limited to resveratrol, proanthocyanidins, and flavanols [162]. An analysis conducted by MI Queipo-Ortuño and colleagues on the impact of red wine extracts on the intestinal microbiota revealed that individuals who consumed red wine over a continuous four-week period exhibited significant increments in the levels of *Enterococcus*, *Prevotella*, *Bifidobacterium*, *Bacteroides uniformis*, *Eggerthella lenta*, and *Blautia coccoides*-*Eubacterium rectale* within their gut. Simultaneously, there was a discernible decrease in the levels of TC, TG, and HDL, a trend closely correlated with the presence of SCFA-producing bacterial species [163]. However, it is crucial to acknowledge that not all plant-derived polyphenols yield positive effects on the regulation of the intestinal microbiota. Researchers found that QR mildly inhibits the growth of *Bifidobacterium* and *Enterococcus* while myricetin suppresses the growth of all LAB without adversely affecting harmful bacteria like *Salmonella* [164,165]. In conclusion, the interactions between polyphenolic compounds and the gut microbiota are multifaceted, encapsulating both positive and negative effects. These complex interactions necessitate further exploration and research for a deeper understanding of their true impact on human health.

### 4.3. Unsaturated Fatty Acids

Improving dietary fat quality by increasing the intake of polyunsaturated fatty acids (PUFAs) while reducing saturated fatty acids (SFAs) significantly decreases serum cholesterol levels [30]. Recent research data illuminate the function of dietary fats as potential modulators of the human gut microbiota composition, with their total amounts and quality acting as pivotal factors in shaping microbial communities in the gut [166,167]. Studies reveal that a higher PUFA intake not only amplifies the total bacterial count within the gut flora but also fosters the proliferation of beneficial bacterial species [168]. Principal sources of PUFAs encompass aquatic species, micro-organisms, algae, and oil crops [169,170,171,172]. Notably, alpha-linolenic acid (ALA), gamma-linolenic acid (GLA), linoleic acid (LA), eicosapentaenoic acid (EPA), and docosahexaenoic acid (DHA) are deemed beneficial for health [172]. It is imperative to acknowledge that PUFAs are categorized into two primary families, ω 6 (n-6) and ω 3 (n-3), with EPA and DHA belonging to omega-3 unsaturated fatty acids [173,174]. Within the human gut, *Lachnospiraceae* and *Bifidobacterium* are identified as beneficial bacteria. The abundance of *Lachnospiraceae* and *Bifidobacterium* negatively correlates with LDL levels [175,176]. These bacterial classes contribute to cholesterol reduction by transforming it into coprostanol [177]. Research conducted by Watson et al. observed a notable increase in the abundance of both *Lachnospiraceae* and *Bifidobacterium* in healthy individuals upon omega-3 PUFA intake [178]. Similarly, a study by Tindall et al. demonstrated that consuming ALA-rich walnuts increases *Lachnospiraceae* [179]. Moreover, studies by Wan et al. found that both EPA and DHA increase *Lachnospiraceae* abundance and positively correlate with the proliferation of various lactic acid-producing bacteria [180]. Research by Li et al. disclosed that Spirulina, rich in LA and GLA, could enhances the abundance of several beneficial bacterial groups in the gut, including *Prevotella*, *Porphyromonadaceae*, *Barnesiella*, and *Parasutterella* [181,182]. Particularly, *Prevotella*, negatively correlated with serum biochemical indicators, promotes bile acid synthesis, further regulating cholesterol metabolism [183].

### 4.4. Phytosterol

Phytosterol (PS) are renowned for their potent cholesterol-lowering effects [184,185]. The primary components of PS include β-sitosterol, stigmasterol, campesterol, and brassicasterol, among others [186]. Studies suggest a daily intake of 2 g of PS can effectively reduce cholesterol levels, particularly TC and LDL-C, by 6–15% [187]. It is noteworthy that lotus seeds, being rich in various bioactive compounds including alkaloids, flavonoid compounds, and PS, are considered excellent food and medicinal sources. Research conducted by Liu et al. revealed that PS in lotus seed cores significantly enhances the abundance of beneficial bacterial phyla in the gut, including *Firmicutes*, *Bacteroides*, *Actinobacteria*, and *Proteobacteria* [188]. *Firmicutes* suppress *Clostridium perfringens* growth, thus maintaining intestinal homeostasis [189]. *Bacteroides* are involved in the metabolism of bile acids and the bioconversion of steroidal compounds. At the same time, certain bacteria within the *Actinobacteria* phylum are known to lower blood sugar and lipid levels. Additionally, soy is a commendable source of PS due to its high content, availability, and safety [190]. Research indicates that upon soy PS intake, there is an increase in the abundance of beneficial gut microbes like *Lactobacillus*, *Oscillibacter*, and *Ackermanella* [191]. Importantly, an increase in *Ackermanella* correlates positively with significant improvements in lipid metabolism and the restoration of colonic mucosal barrier functions [192].

### 4.5. Dicaffeoylquinic Acid

Kuding Tea (KDC), popular in China and Southeast Asian nations like Singapore and Malaysia, is recognized as a functional tea beverage known for its multiple pharmacological activities including dispelling wind-heat, quenching thirst, eliminating phlegm, and boosting alertness [193]. KDC is rich in caffeoylquinic acid derivatives with antioxidant activities, such as 3-CQA, 5-CQA, 3,4-diCQA, 3,5-diCQA, and 4,5-diCQA [194]. Research conducted by Xie et al. discovered that dicaffeoylquinic acid (DCQA) in Kuding tea modulates cholesterol metabolism in mice and promotes the growth of beneficial gut microbes like *Bifidobacterium* and *Akkermansia muciniphila* [31]. These microbial populations’ alterations subsequently influence microbial community functions, including bile acid biosynthesis. Notably, the genus *Odoribacter*, belonging to the *Porphyromonadaceae* family, is identified as a primary producer of Ac, Pr, and Bu, which are SCFAs proven to lower cholesterol levels effectively [195,196]. DCQA adjusts the relative abundance of gut microbes like *Odoribacter*, *Prevotella*, *Bacteroides*, *Parasutterella*, and *Lachnospiraceae*, effectively ameliorating gut dysbiosis [194]. Furthermore, DCQA alters the functional characteristics of the gut microbial community, providing a potential mechanism foundation for maintaining gut health and regulating cholesterol levels.

## 5. Conclusions

Gut microbiota dysbiosis is a risk factor in the pathophysiological processes related to cholesterol-associated diseases, constituting a subtle and potential mechanism of disease onset. This mechanism, directly or indirectly, influences human health. Notably, in cardiovascular diseases, abnormal cholesterol levels facilitate the formation and development of atherosclerotic plaques by inducing the generation of oxidized LDL. Increasing evidence suggests that a healthy gut microbiota engages in cholesterol reduction through various pathways, making it imperative to explore the precise mechanisms by which it achieves this. The conduction and findings of clinical trials will offer deeper insights into treating the cardiovascular diseases induced by high blood cholesterol. Furthermore, the interaction between natural functional ingredients and the cholesterol-lowering actions of the gut microbiota also represents a significant focus of research. This focus is poised to profoundly impact the development of novel therapeutic strategies for drug treatment. In summary, a deeper understanding of the mechanisms through which the gut microbiota reduces cholesterol is scientifically essential and opens a new avenue for the prevention of cardiovascular diseases. Prospective studies should further deepen the understanding of the connection between the gut microbiota and cholesterol reduction while exploring and identifying more effective prevention and treatment strategies.

## Figures and Tables

**Figure 1 foods-12-04308-f001:**
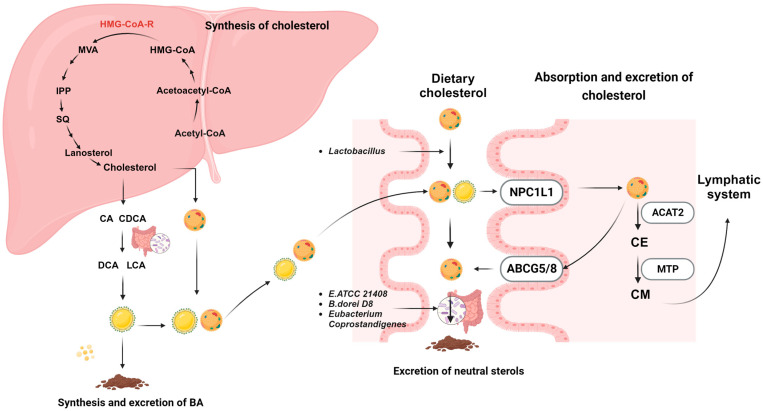
Absorption, biosynthesis, and metabolic process of cholesterol. ABCG5/8, ATP binding transporter protein 5/8; ACAT2, Acyl-coenzyme: cholesterol acyltransferases 2; BA, Bile acid; CA, Cholic acid; CDCA, Chenodeoxycholic acid; CM, Chylomicron; DCA, Deoxycholic acid; HMG-CoA-R, HMG-CoA reductase; HMG-CoA-S, HMG-CoA synthase; IPP, Isopentoyl disphosphate; LCA, Lithocholic acid; MTP, Microsomal triglyceride transfer protein; MVA, Mevalonate; NPC1L1, Niemann-Pick C1-like 1 protein; SQ, squalene. 
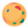
, cholesterol; 
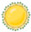
, bile acids.

**Figure 2 foods-12-04308-f002:**
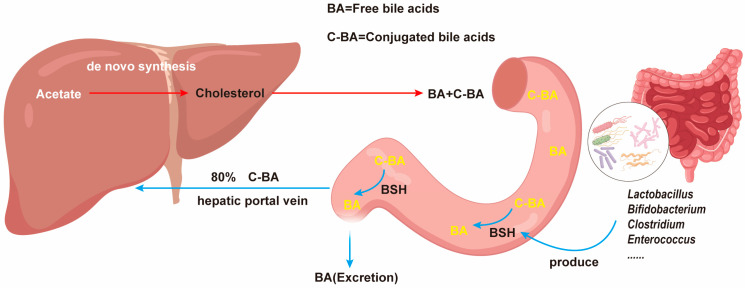
BSH, an advantageous enzyme synthesized by the gut microbiota, participates in cholesterol metabolism through the hydrolysis of conjugated bile acids.

**Figure 3 foods-12-04308-f003:**
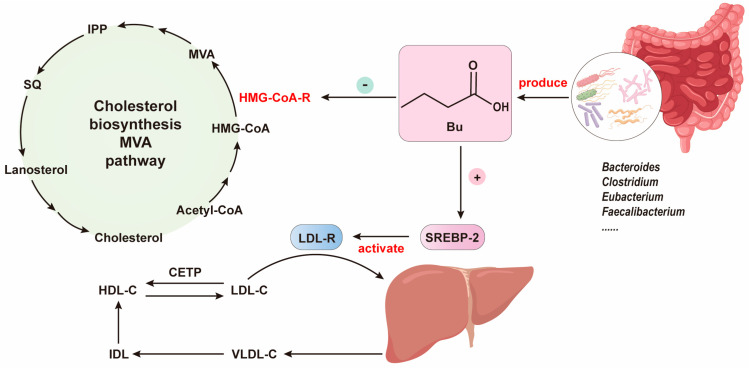
The role of SCFAs, metabolic products of gut microbiota, in cholesterol metabolism. Bu, Butyrate; CETP, Cholesteryl ester transfer protein; HDL-C, High-density lipoproteins cholesterol; HMG-CoA-R, HMG-CoA reductase; IDL, Intermediate-density lipoproteins; IPP, Isopentoyl disphosphate; LDL-C, Low-density lipoprotein cholesterol; LDLR, LDL receptors; MVA, Mevalonate; SQ, Squalene; SREBP2, Sterol-regulatory element binding protein-2; VLDL-C, Very-low-density lipoproteins cholesterol.

**Figure 4 foods-12-04308-f004:**
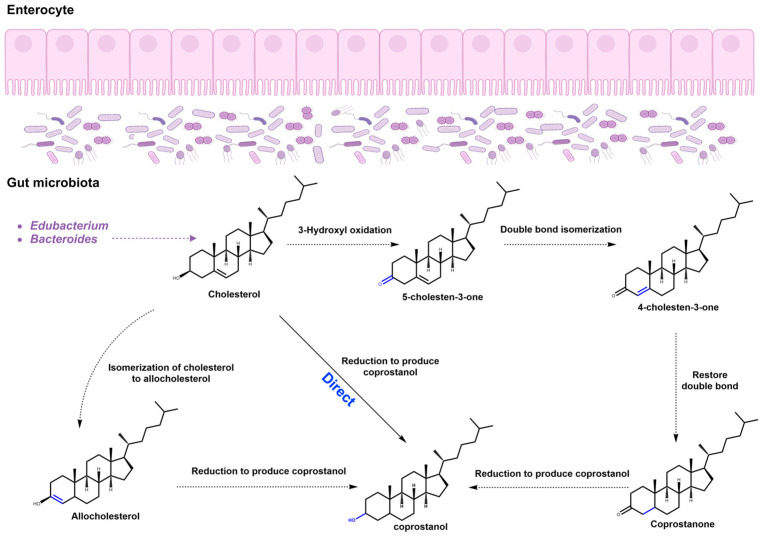
Metabolic pathway of cholesterol conversion to coprostanol.

## Data Availability

The data are openly available in a public repository. To collect the articles cited in this review, we conducted a comprehensive literature search in databases such as PubMed, Web of Science, and Google Scholar. We used keywords related to our research topic, like ‘cholesterol metabolism’ and ‘gut microbiota’, for the search. During the search process, we did not set any specific time constraints to ensure the inclusion of as many relevant studies as possible. Our only criterion for selection was language, considering only articles written in English. This approach was taken to ensure that we could accurately understand and analyze the content of the selected literature. No other specific inclusion or exclusion criteria were set, allowing us to comprehensively evaluate all relevant research, thereby ensuring the breadth and depth of the review.

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
