# Peer review of "Effect of Gut Microbiota on Blood Cholesterol: A Review on Mechanisms"

_foods, 2023, doi:10.3390/foods12234308_

Round 1

Reviewer 1 Report

Comments and Suggestions for Authors

In the manuscript "Effect of Gut Microbiota on Blood Cholesterol: A Review on Mechanisms" authors describe different mechanisms by which intestinal microbiota affects blood cholesterol levels. 

The review is very interesting and fits within the scope of the journal. 

some minor comments:

Abstract, line 17: delete a space in "G ut"

line 95: please put "Ilex kudingcha" in italic letter. 

Line 135: intestines? small and large intestines?

Line 207: lactic acid bacteria should not be written in italics. 

Line 223: human intestine? large intestine?

Line 302: "Lu" is Leucine? or isobutyrate (see line 55).

Lines 305-338: please check the new names of Lactobacillus species. Apply them when neccesary (if the cited paper is new and posterior to the new classification). 

Line 419: Thetaiotaomicron Bacteroides? or Bacteroides thetaiotaomicron?

Line 422: delete "-" 

Lines 426, 428 and 439 and all over the manuscript : put the names of genera and species in italics.

Author Response

  1. Abstract, line 17: delete a space in "G ut"; line 95: please put "Ilex kudingcha" in italic letter; Line 135: intestines? small and large intestines? Line 207: lactic acid bacteria should not be written in italics; Line 223: human intestine? large intestine?

Response: Thanks for your suggestion. As suggested, the space in “G ut” was removed, please see line 17; "Ilex kudingcha" was changed to "Ilex kudingcha", please see line 95; “intestines” was corrected to “small intestines”, please see line 136; “lactic acid bacteria” was changed to "lactic acid bacteria", please see line 210; “human intestine” was corrected to “large intestine”, please see line 226.

  1. Line 302: "Lu" is Leucine? or isobutyrate (see line 55).

Response: Thanks for your suggestion. As suggested, “Leucine (Lu)” was changed to “isobutyrate”, please see line 307.

  1. Lines 305-338: please check the new names of Lactobacillus species. Apply them when neccesary (if the cited paper is new and posterior to the new classification).

Response: Thanks for your suggestion. As suggested, We reviewed the literature information and changed "Lactobacillus" to "Lactic Acid Bacteria", please see line 311.

  1. Line 419: Thetaiotaomicron Bacteroides? or Bacteroides thetaiotaomicron?

Response: Thanks for your suggestion. As suggested, “Thetaiotaomicron Bacteroides” was changed to “Bacteroides thetaiotaomicron”, please see line 424.

  1. Line 422: delete "-"; Lines 426, 428 and 439 and all over the manuscript : put the names of genera and species in italics.

Response: Thanks for your suggestion. As suggested, we removed the “-” on Line 427, and italicized all genus and species names throughout the manuscript.

Reviewer 2 Report

Comments and Suggestions for Authors

The manuscript has been written well. However, I have two observations for the improvement of the manuscript:

1) The authors should show the mechanism(s) of gut microbiota (for example, absorption, biosynthesis inhibition and all other mechanisms) in a Figure. 

2) The authors should include the article search strategy and databases for the collection of articles in this review

Author Response

  1. The authors should show the mechanism(s) of gut microbiota (for example, absorption, biosynthesis inhibition and all other mechanisms) in a Figure.

Response: Thanks for your suggestion. It is important to show the interaction between gut microbiota and cholesterol metabolism mechanism. In fact, existing figures have shown the relevant microbiota involved in cholesterol biosynthesis and excretion. Please see Figure 1, Figure 2 and Figure 3.

  1. The authors should include the article search strategy and databases for the collection of articles in this review.

Response: Thanks for your suggestion. As suggested, the article search strategy and databases have been added in this review. Please see line 567-575 “To collect the articles cited in this review, we conducted a comprehensive literature search in databases such as PubMed, Web of Science, and Google Scholar. We used keywords related to our research topic, like 'cholesterol metabolism' and 'gut microbiota', for the search. During the search process, we did not set any specific time constraints to ensure the inclusion of as many relevant studies as possible. Our only criterion for selection was language, considering only articles written in English. This approach was taken to ensure that we could accurately understand and analyze the content of the selected literature. No other specific inclusion or exclusion criteria were set, allowing us to comprehensively evaluate all relevant research, thereby ensuring the breadth and depth of the review.”

Reviewer 3 Report

Comments and Suggestions for Authors

There are several exhaustive review articles published on this topic already. I am not sure what new information the authors have shared in this review article or how their article is different from others?

Vourakis M, Mayer G, Rousseau G. The Role of Gut Microbiota on Cholesterol Metabolism in Atherosclerosis. Int J Mol Sci. 2021 Jul 28;22(15):8074. doi: 10.3390/ijms22158074. PMID: 34360839; PMCID: PMC8347163.

Jia B, Zou Y, Han X, Bae JW, Jeon CO. Gut microbiome-mediated mechanisms for reducing cholesterol levels: implications for ameliorating cardiovascular disease. Trends Microbiol. 2023 Jan;31(1):76-91. doi: 10.1016/j.tim.2022.08.003. Epub 2022 Aug 22. PMID: 36008191.

Comments on the Quality of English Language

Moderate English language copyediting will suffice...

Author Response

There are several exhaustive review articles published on this topic already. I am not sure what new information the authors have shared in this review article or how their article is different from others?

Response: Thanks for your suggestion. As suggested, We have compared these articles with this article.

"The Role of Gut Microbiota on Cholesterol Metabolism in Atherosclerosis" explores the impact of microbial metabolites such as bile acids, trimethylamine N-oxide, and short-chain fatty acids on cholesterol metabolism. This paper delves into how these metabolites are modulated through nutritional and pharmacological interventions and their subsequent influence on cardiovascular health.

"Gut microbiome-mediated mechanisms for reducing cholesterol levels: implications for ameliorating cardiovascular disease" focuses on the application of probiotics in reducing cholesterol levels and discusses the potential of next-generation probiotics, such as Akkermansia and Bacteroides spp., in the prevention and treatment of cardiovascular diseases.

Compared to the other two articles, this article provides a comprehensive exploration of multiple mechanisms through which the gut microbiota impacts cholesterol metabolism. It examines five distinct methods: altering the ratio of free to conjugated bile acids, fermenting dietary fiber to produce SCFAs, regulating cholesterol metabolism-related gene expression through metabolites, converting cholesterol to coprostanol with low absorption rates, and reducing cholesterol by inhibiting the production of lipopolysaccharide. Additionally, this paper highlights the role of natural constituents in food, such as polysaccharides, polyphenolic compounds, unsaturated fatty acids, phytosterols, and dicaffeoylquinic acid, in cholesterol regulation through their interactions with the gut microbiota. Its innovation lies in providing a comprehensive perspective on how the gut microbiota influences cholesterol metabolism through various biological pathways and interactions with dietary components.

Reviewer 4 Report

Comments and Suggestions for Authors

The authors summary five possible mechanisms of gut microbiota influence cholesterol metabolism. In addition, five types of foods constituents (polysaccharides, polyphenolic entities, polyunsaturated fatty acids, phytosterols, and dicaffeoylquinic acid) with potential roles of gut microbiota in cholesterol regulation were also discussed.

The following question arises.

  文本框: 1. The five possible mechanisms described in the article shall be consistent. For example, 1) Gut microbiota changes the ratio of free bile acids to conjugated bile acids with the former being eliminated into feces and the latter being reabsorbed back into the liver;

à Gut microbiota modify conjugated bile acids.

2.  In the article, too many old references are cited. The article is full of repetition and redundancy. All the figures shall be labeled with the microorganisms participated in the mechanism.

3.  Section 4

Natural Functional Constituents Influencing Gut Microbiota in the Regulation of Cholesterol Metabolism

à The authors shall list the natural functional constituents in a new table. In addition, the possible mechanisms shall be discusses and the microorganisms altered blood lipid composition shall be listed.

Line 17: G utà Gut

Line 54: hexanoate (Ca)à caproate (Ca)

Figure 1:

The figure resolution is not good enough. Lanosterol is absent in the figure. Figure legend shall be rewritten.

Line 162-187: need another new figure

Line 182: fecal-neutral sterols is not shown in Figure 1. Line 188: amdà and

Line 194~199: Four possible mechanisms are described here. LPS in cholesterol metabolism is absent here.

Line 200~268:

The article described here is full of repetition and redundancy.

A new figure shall be added in article. In addition, the microorganisms participated in the conversion of bile acid shall be added in the new figure.

Figure 2:

The figure title shall not describe BSH only. Figure title and legend shall be rewritten.

Figure 3: Figure legend

HDL-C, High-density lipoproteinsà High-density lipoproteins cholesterol

VLDL-C, Very-low-density lipoproteinsà Very-low-density lipoproteins cholesterol

Line 281:

Sterol Regulatory Element-SREBP-2 àSterol-regulatory element binding protein-2

Line 302:

leucine (Lu) àisobutyrate (Lu)

Why is the abbreviation of isobutyrate Lu?                                                                                                                 

Line 355:

cholesterol is oxidized to intermediary 4-cholesten-3-one under the influence of cholesterol oxidase; subsequently, 4-cholesten-3-one undergoes isomerization to form 5-alpha-cholestan-3-one

àbut in the figure 4, cholesterol is oxidized to 5-alpha-cholestan-3-one, then isomerized to 4-cholesten-3-one. The authors shall confirm its correctness.

Line 367:

Recent studies have shown

à but the references cited here are the papers published in 1953 and 1973.

Line 377:

Figure 4 shall be added the microorganisms participated in the conversion to coprostanol.

Line 414: Indigetible à Indigestible

Line 419:

Thetaiotaomicron Bacteroides à Bacteroides thetaiotaomicron

Line 421: What is the halli bacteria”?

Line 426: Enteromorphaà red algae. Line 430: Aalginate à Alginate

Line 447: X Tzounis et al. à Tzounis et al.

Line 447: Blautia coccoides– Eubacterium rectale

à Blautia coccoides-Eubacterium rectale group

Line 462: Researchers A Duda-Chodak and J

Line 496: Plant sterol shall be unified as phytosterol in the article. Line 514: 4.4 shall be 4.5.

Line 979: graphics Fecosterol à coprostanol Cholic acid à bile acids

Comments on the Quality of English Language

The authors summary five possible mechanisms of gut microbiota influence cholesterol metabolism. In addition, five types of foods constituents (polysaccharides, polyphenolic entities, polyunsaturated fatty acids, phytosterols, and dicaffeoylquinic acid) with potential roles of gut microbiota in cholesterol regulation were also discussed.

The following question arises.

  文本框: 1. The five possible mechanisms described in the article shall be consistent. For example, 1) Gut microbiota changes the ratio of free bile acids to conjugated bile acids with the former being eliminated into feces and the latter being reabsorbed back into the liver;

à Gut microbiota modify conjugated bile acids.

2.  In the article, too many old references are cited. The article is full of repetition and redundancy. All the figures shall be labeled with the microorganisms participated in the mechanism.

3.  Section 4

Natural Functional Constituents Influencing Gut Microbiota in the Regulation of Cholesterol Metabolism

à The authors shall list the natural functional constituents in a new table. In addition, the possible mechanisms shall be discusses and the microorganisms altered blood lipid composition shall be listed.

Line 17: G utà Gut

Line 54: hexanoate (Ca)à caproate (Ca)

Figure 1:

The figure resolution is not good enough. Lanosterol is absent in the figure. Figure legend shall be rewritten.

Line 162-187: need another new figure

Line 182: fecal-neutral sterols is not shown in Figure 1. Line 188: amdà and

Line 194~199: Four possible mechanisms are described here. LPS in cholesterol metabolism is absent here.

Line 200~268:

The article described here is full of repetition and redundancy.

A new figure shall be added in article. In addition, the microorganisms participated in the conversion of bile acid shall be added in the new figure.

Figure 2:

The figure title shall not describe BSH only. Figure title and legend shall be rewritten.

Figure 3: Figure legend

HDL-C, High-density lipoproteinsà High-density lipoproteins cholesterol

VLDL-C, Very-low-density lipoproteinsà Very-low-density lipoproteins cholesterol

Line 281:

Sterol Regulatory Element-SREBP-2 àSterol-regulatory element binding protein-2

Line 302:

leucine (Lu) àisobutyrate (Lu)

Why is the abbreviation of isobutyrate Lu?                                                                                                                 

Line 355:

cholesterol is oxidized to intermediary 4-cholesten-3-one under the influence of cholesterol oxidase; subsequently, 4-cholesten-3-one undergoes isomerization to form 5-alpha-cholestan-3-one

àbut in the figure 4, cholesterol is oxidized to 5-alpha-cholestan-3-one, then isomerized to 4-cholesten-3-one. The authors shall confirm its correctness.

Line 367:

Recent studies have shown

à but the references cited here are the papers published in 1953 and 1973.

Line 377:

Figure 4 shall be added the microorganisms participated in the conversion to coprostanol.

Line 414: Indigetible à Indigestible

Line 419:

Thetaiotaomicron Bacteroides à Bacteroides thetaiotaomicron

Line 421: What is the halli bacteria”?

Line 426: Enteromorphaà red algae. Line 430: Aalginate à Alginate

Line 447: X Tzounis et al. à Tzounis et al.

Line 447: Blautia coccoides– Eubacterium rectale

à Blautia coccoides-Eubacterium rectale group

Line 462: Researchers A Duda-Chodak and J

Line 496: Plant sterol shall be unified as phytosterol in the article. Line 514: 4.4 shall be 4.5.

Line 979: graphics Fecosterol à coprostanol Cholic acid à bile acids

Author Response

  1. The five possible mechanisms described in the article shall be consistent. For example, 1) Gut microbiota changes the ratio of free bile acids to conjugated bile acids with the former being eliminated into feces and the latter being reabsorbed back into the liver;

Response:

Thanks for your suggestion. In this review, we delve into five possible mechanisms by which gut microbiota may influence cholesterol metabolism: 1) Gut microbiota changes the ratio of free bile acids to conjugated bile acids with the former being eliminated into feces and the latter being reabsorbed back into the liver; 2) Gut microbiota can ferment dietary fiber to produce short-chain fatty acids (SCFAs) which are absorbed and reach the liver where SCFAs inhibit the cholesterol synthesis; 3) Gut microbiota can regulate the expression of some genes related to cholesterol metabolism through their metabolites; 4) Gut microbiota can convert cholesterol to coprostanol with the latter having a very low absorption rate; and 5) Gut microbiota could reduce blood cholesterol by inhibiting the production of lipopolysaccharide (LPS), which increases cholesterol synthesis and raises blood cholesterol. Actually, these five possible mechanisms described in the article was consistent. After careful consideration, we decided to maintain the original descriptions of the mechanisms presented in article.

  1. In the article, too many old references are cited. The article is full of repetition and redundancy. All the figures shall be labeled with the microorganisms participated in the mechanism.

Response: Thanks for your suggestion. As suggested, all the figures was labeled with the microorganisms participated in the mechanism. However, the cited old references are crucial for the context, and the detailed description is essential for emphasizing the key points. Hence, we prefer to retain these old references and  description in our manuscript.

  1. Natural Functional Constituents Influencing Gut Microbiota in the Regulation of Cholesterol Metabolism. The authors shall list the natural functional constituents in a new table. In addition, the possible mechanisms shall be discusses and the microorganisms altered blood lipid composition shall be listed.

Response: Thanks for your suggestion. In this study, we focused on five possible mechanisms by which gut microbiota affect cholesterol metabolism. Over-exposing specific functional components and detailed mechanisms may distract from our research focus and make the content too redundant. Therefore, we kindly request to retain the current format of the manuscript.

  1. Line 17: G ut→Gut; Line 54: hexanoate (Ca)→caproate (Ca); Line 188: amd→and.

Response: Thanks for your suggestion. As suggested, the space in “G ut” was removed, please see line 17; “hexanoate (Ca)” was changed to “caproate (Ca)”, please see line 54 ; “amd” was changed to “and”, please see line 172.

  1. Figure 1: The figure resolution is not good enough. Lanosterol is absent in the figure. Figure legend shall be rewritten.

Response: Thanks for your suggestion. As suggested, the figure resolution was improved to 600 DPI and lanosterol was added in Figure 1.

  1. Line 162-187: need another new figure.

Response: Thanks for your suggestion. Due to the excessive amount of figures in this manuscript, we prefer to use words to describe the process of gut microbiota converting primary bile acids into secondary bile acids.

  1. Line 182: fecal-neutral sterols is not shown in Figure 1.

Response: Thanks for your suggestion. The right side of Figure 1 illustrates the mechanisms of cholesterol absorption and excretion. Cholesterol is absorbed in the intestines, and a portion that is not absorbed is modified by bacteria into neutral sterols, which are then excreted with feces. A more detailed diagram of the conversion of cholesterol into neutral sterols is presented in Figure 4.

  1. Line 194~199: Four possible mechanisms are described here. LPS in cholesterol metabolism is absent here.

Response: Thanks for your suggestion. As suggested, LPS was added in cholesterol metabolism. Please see line 201 “and influencing the production of LPS, which affects cholesterol levels”.

  1. Line 200~268: The article described here is full of repetition and redundancy. A new figure shall be added in article. In addition, the microorganisms participated in the conversion of bile acid shall be added in the new figure.

Response: Thanks for your suggestion. The process of microorganisms participated in the conversion of bile acid was described in Figure 2 and line 163-174.

  1. Figure 2: The figure title shall not describe BSH only. Figure title and legend shall be rewritten.

Response: Thanks for your suggestion. After careful consideration, we decided to retain the current figure title and legend, because the current title succinctly summarizes the main focus of the figure, which is the role of BSH.

  1. Figure 3: Figure legend, HDL-C, High-density lipoproteins→High-density lipoproteins cholesterol. VLDL-C, Very-low-density lipoproteins→Very-low-density lipoproteins cholesterol.

Response: Thanks for your suggestion. In    Figure 3, “HDL-C, High-density lipoproteins” was changed to “High-density lipoproteins cholesterol”; “VLDL-C, Very-low-density lipoproteins” was changed to “Very-low-density lipoproteins cholesterol”.

  1. Line 281:Sterol Regulatory Element-SREBP-2→Sterol-regulatory element binding protein-2; Line 302:leucine (Lu)→isobutyrate (Lu); Why is the abbreviation of isobutyrate Lu?

Response: Thanks for your suggestion. As suggested, “Sterol Regulatory Element-SREBP-2” was changed to “Sterol-regulatory element binding protein-2”, please see line 285; We reviewed relevant materials and found that isobutyrate does not have a universally recognized abbreviation, thus we changed all instances of "isobutyrate (Lu)" in the text to "isobutyrate".

  1. Line 355: cholesterol is oxidized to intermediary 4-cholesten-3-one under the influence of cholesterol oxidase; subsequently, 4-cholesten-3-one undergoes isomerization to form 5-alpha-cholestan-3-one. but in the figure 4, cholesterol is oxidized to 5-alpha-cholestan-3-one, then isomerized to 4-cholesten-3-one. The authors shall confirm its correctness.

Response: Thanks for your suggestion. As suggested, We reconfirmed the steps of cholesterol conversion to coprostanol and revised the original text, please see line 360-362 "cholesterol is oxidized to intermediary 5-alpha-cholestan-3-one under the influence of cholesterol oxidase; subsequently, 5-alpha-cholestan-3-one undergoes isomerization to form 4-cholesten-3-one."

  1. Line 367: Recent studies have shown→but the references cited here are the papers published in 1953 and 1973.

Response: Thanks for your suggestion. As suggested, we removed the phrase “recent studies” to more accurately reflect the year of these references.

  1. Line 377: Figure 4 shall be added the microorganisms participated in the conversion to coprostanol.

Response: Thanks for your suggestion. As suggested, we added the microorganisms (Eubacterium and Bacteroides) involved in the conversion of cholesterol to coprostanol in Figure 4.

  1. Line 414: Indigetible→Indigestible; Line 419: Thetaiotaomicron Bacteroides→Bacteroides thetaiotaomicron; Line 421: What is the “halli bacteria”?

Response: Thanks for your suggestion. As suggested, “Indigetible” was changed to “Indigestible”, please see line 419; “Thetaiotaomicron Bacteroides” was changed to “Bacteroides thetaiotaomicron”, please see line 424; We verified the reference and confirmed that Eubacterium rectale convert Pr and Ac into butyrate. Therefore, “halli bacteria” was changed to “Eubacterium rectale”, please see line 426.

  1. Line 426: Enteromorpha→red algae; Line 430: Aalginate→Alginate; Line 447: X Tzounis et al.→Tzounis et al.; Line 447: Blautia coccoides– Eubacterium rectale →Blautia coccoides-Eubacterium rectale group; Line 462: Researchers A Duda-Chodak and J; Line 496: Plant sterol shall be unified as phytosterol in the article; Line 514: 4.4 shall be 4.5; Line 979: graphics-Fecosterol→coprostanol, Cholic acid→bile acids.

Response: Thanks for your suggestion. As suggested, “Enteromorpha” was changed to “red algae”, please see line 431; “Blautia coccoides– Eubacterium rectale” was changed to “Blautia coccoides-Eubacterium rectale group” , please see line 453; “Researchers A Duda-Chodak and J” was changed to “Researchers” , please see line 467; throughout the text, "Plant sterol" was changed to "phytosterol"; in the graphics, “Fecosterol” was changed to “coprostanol” and “cholic acid was changed to “bile acids” , please see line 995.

Round 2

Reviewer 4 Report

Comments and Suggestions for Authors

The paper can be accepted without any further changes.